# Perceived Occupational Noise Exposure and Depression in Young Finnish Adults

**DOI:** 10.3390/ijerph20064850

**Published:** 2023-03-09

**Authors:** Marja Heinonen-Guzejev, Alyce M. Whipp, Zhiyang Wang, Anu Ranjit, Teemu Palviainen, Irene van Kamp, Jaakko Kaprio

**Affiliations:** 1Clinicum, Department of Public Health, University of Helsinki, FI-00014 Helsinki, Finland; 2Institute for Molecular Medicine Finland (FIMM), University of Helsinki, FI-00014 Helsinki, Finland; 3Centre for Sustainability, Environment and Health, National Institute for Public Health and the Environment in the Netherlands (RIVM), 3720 BA Bilthoven, The Netherlands

**Keywords:** occupational noise, noise sensitivity, depressive symptoms, young adults

## Abstract

We investigated the association between perceived occupational noise exposure and depressive symptoms in young Finnish adults and whether noise sensitivity moderates this association. This study was based on an ongoing longitudinal twin study. We included those who had been working daily (*n* = 521) or weekly (*n* = 245) during the past 12 months (mean age 22.4, SD 0.7, 53% female). We asked about occupational noise exposure at age 22 and assessed depressive symptoms using the General Behavior Inventory (GBI) at age 17 and 22. Noise sensitivity and covariates were used in linear regression models. Perceived daily occupational noise exposure was associated, as a statistically independent main effect with depressive symptoms at age 22 (beta 1.19; 95% CI 0.09, 2.29) among all, and separately for females (beta 2.22; 95% CI 0.34, 4.09) but not males (beta 0.22; 95% CI −1.08, 1.52). Noise sensitivity was independently associated with depressive symptoms among all (beta 1.35; 95% CI 0.54, 2.17), and separately for males (beta 1.96; 95% CI 0.68, 3.24) but not females (beta 1.05; 95 % CI −0.04, 2.13). Noise sensitivity was independent of perceived occupational noise exposure. Pre-existing depressive symptoms at age 17 were predictive of perceived occupational noise exposure, suggesting complex interactions of noise and depression.

## 1. Introduction

According to the World Health Organization (WHO), depression occurs among 1.1% of adolescents aged 10–14 years, and 2.8% of 15–19-year-olds [1]. It is estimated that, in adulthood, twice as many females as males are depressed [2]. In children, the prevalence of depression is below 1%, with no sex differences. The prevalence rises throughout adolescence [2]. The sex differences first emerge in the age range of 11–14 [3]. The main risk factors for depression in adolescence are family history of depression and exposure to psychosocial stress [4]. Genetic and developmental factors, psychosocial adversity, and sex hormones interact and increase the risk of depression [4]. Other risk factors for depression in youth are affect, cognition, and behavior [5]. Studies have shown that depressed adolescents maintain longer depressive affective states [6]. Cognitive mechanisms at least partially mediate the relationship between parenting and depressive symptoms [7]. Parents can model depressive behavior or they may fail to model problem-solving, coping skills or affect regulation [5]. All of these risk factors at the individual and broader contextual levels interact, leading to the development of depression [5].

Exposure to physical environmental agents and perinatal, familial and social factors can have substantial effects on human development. The exposome is considered to consist of a general external domain (e.g., urbanity, social capital, and stress), specific external domains (e.g., diet, smoking, infections, pollutants, and chemical exposures), and internal domains (e.g., gut microbiota, oxidative stress, and metabolism) [8,9]. This study is part of the Equal-Life project, which contributes to the development of the exposome concept by integrating the internal, physical and social exposomes. We distinguish between the external exposome, which is subdivided into the physical exposome (e.g., environmental indoor and outdoor quality, such as noise exposure), the social exposome (e.g., societal context, socioeconomic, social, and psychosocial factors), and the internal exposome [10]. The general exposome factor has been associated with psychopathology [11]. Genome–exposome interactions seem to contribute substantially to youth mental health symptoms [12].

Noise is an unwanted sound that may adversely affect the health and well-being of individuals. It is an environmental stressor, which can also affect mental health [13,14]. There are several studies on the effects of environmental noise on mental health and, according to a systematic review by Clark et al. (2020), there is low quality evidence that road traffic noise may have harmful effects on interview-measured depression and anxiety as well as on medication use [15].

Noise annoyance is a person’s individual adverse reaction to noise [16]. According to recent studies, noise annoyance can be a mediator between noise exposure and mental ill-health [17], especially depression [18]. Noise sensitive individuals are more affected by noise and noise sensitivity predicts noise annoyance [19]. Noise sensitivity aggregates in families and probably has a genetic component. The estimate of heritability of noise sensitivity is 36% [20]. Noise sensitivity can be a moderator of noise exposure on psychological ill-health [17,21,22].

Studies on occupational noise have mainly investigated the auditory effects of noise and we have less information on occupational noise and other health outcomes, particularly mental health. We are missing solid evidence from young people entering the workforce. Noise exposure at work is one of the risk factors for a poorer quality of life [23]. Occupational noise annoyance has been significantly related to mental health, including depressive symptoms and suicidal ideation [24]. In a study of the effects of noise on the workload and stress levels of operating room staff, anxiety and workload scores were positively correlated with noise levels [25]. Noise is also one of the occupational hazards affecting job stress [26]. In a study on workplace noise in an emergency department, the performance of mental tasks was maintained during noise exposure, but noise exposure was associated with significant degrees of self-reported distress [27]. In a study by Sjödin et al. (2012) on noise and stress effects on preschool personnel, the adverse effects were studied using questionnaires and saliva cortisol samples. To measure the subjective stress and its relation to the psychosocial work conditions, the Stress–Energy adjective checklist was used. Stress and energy output were pronounced. About 30% of the staff experienced strong burnout symptoms that were associated with reduced sleep quality and increased morning sleepiness. Mental recovery after work was low, which was indicated by stress levels remaining high after work. Cortisol levels supported the conclusion regarding pronounced elevated daily stress levels [28]. In a study by van Dijk et al. (1987), about two-thirds of workers reported noise annoyance. Consistent positive relationships were found between symptoms of stress and noise annoyance. Mentally stressful tasks were most affected by noise [29].

This study was prepared as part of the Equal-Life project, which studies long-term effects of early environmental exposures during childhood and adolescence on mental health and cognition. We aimed to characterize the effects of perceived occupational noise exposure on depressive symptoms among young adults from the longitudinal twin study (FinnTwin12 cohort). We also assessed pre-existing depressive symptoms measured at age 14 and 17 for their association with future perceived noise exposure. Noise exposure at work at that age would be minimal or absent, thus providing clues as to the direction of effects. Those with earlier depressive symptoms could, e.g., report the same noise levels as louder than those without depressive symptoms, or the association of prior depression with occupational noise in the future could be one manifestation of a healthy worker selection effect [30,31]. A secondary aim of this study was to establish whether noise sensitivity moderates an association between perceived noise exposure and depressive symptoms.

## 2. Materials and Methods

This study was based on FinnTwin12 (FT12), an ongoing longitudinal twin study launched in 1994 to investigate the developmental genetic epidemiology of health-related behaviors. In 1994, 5600 twins were enrolled and their families selected from nation-wide birth cohorts born from 1983–1987. In the first phase, questionnaires were sent to twins, their parents and teachers when the twins were 11–12 years old, with two follow-ups at ages 14 and 17.5 years [32,33]. Twins from 935 families were invited to in-depth assessments at age 14, including psychiatric interviews [34]. At age 14, 1852 twins participated and they were contacted again for participation as young adults (wave 4). In the 2006–2009 wave 4 study, data collection involved conducting structured psychiatric interviews at an average age of 21.9 (SD 0.8, range 21–26) years, with neuropsychological assessments and collection of DNA and serum samples (for basic biochemistry and metabolomics) in addition to completing questionnaires including items on work, work exposures and work conditions. A total of 1347 twins completed interviews at age 22, while 1299 completed questionnaires related to work [35].

Inclusion criteria: We included those who had been working during the past 12 months (*n* = 766, 53% female, age range 20–26 years). Work during the past 12 months was assessed using the question “During the last 12 months have you been at work?” The response choices were as follows: 1 “daily”, 2 “weekly”, 3 “monthly”, 4 “temporary work”, 5 “I have not worked at all or it has been very occasional”. Of these participants, 521 had worked daily and 245 weekly. We also asked about studying and working using the question “Do you go to school or are you studying at the moment?” The response choices were as follows: 1 “No, I am not studying anywhere”, 2 “I’m studying, not working at the same time”, 3 “I’m studying and also working ______ hours per week”. The other 533 participants had not worked or worked infrequently for several reasons (such as full-time study, parental leave, military service or unemployment).

Exposure data: Perceived occupational noise exposure at age 22 (wave 4) was defined by asking about noise exposure at work. We used the question “How often is there such a loud noise at your work that you cannot hear normal speech within a meter’s distance?” The response choices were 1 “daily”, 2 “weekly”, 3 “occasionally”, 4 “never.” All four categories were used in the analyses.

Covariates: We based our selection of covariates on their availability in the FT12 study and their use in previous studies of relevance for depression [36,37]. We included age, sex, education, economic situation, work conditions, and smoking status as relevant covariates. Two noise-specific variables were included, i.e., noise sensitivity and noise annoyance.

Noise annoyance is the most prevalent community response in populations exposed to environmental noise [38]. Annoyance is related to noise level, but noise sensitivity is not [39,40]. EU Directive 2002/49/EC (1) recommends evaluating environmental noise exposures on the basis of estimated noise annoyance [41]. Noise annoyance is assessed at the level of populations using a questionnaire. Noise annoyance was studied using the question “Do the following factors annoy you at work: noise?” The response alternatives were 1 “no or does not bother”, 2 “bothers somewhat”, 3 “bothers quite a lot”, 4 “bothers very much”. Participants answering “no or does not bother” were classified as not reporting noise annoyance. Participants answering “bothers somewhat” were classified as reporting some annoyance. Participants answering “bothers quite a lot” were classified as reporting moderate annoyance. Participants answering “bothers very much” were classified as reporting high annoyance.

Noise sensitivity refers to physiological and psychological internal states of an individual that increase the degree of reactivity to noise in general [42]. Noise sensitive individuals pay more attention to noise than less sensitive ones. They have higher levels of tonic physiological arousal and more startle responses to noise. In depressed patients, noise sensitivity scores have been shown to be considerably higher than in control participants [19]. Noise sensitivity is a potential moderator of an association between noise exposure and depression. Noise sensitivity was studied using the statement “An unexpected noise makes me jump and startle.” The response choices were “Strongly disagree”, “Disagree”, “Agree” or “Strongly agree”. In the analyses, four classes of noise sensitivity were re-categorized into two. Persons with noise sensitivity were those answering “Agree” or “Strongly agree” and persons without noise sensitivity were those answering “Strongly disagree” or “Disagree”.

Work conditions were studied using the Karasek Job Control, Demand and Support Scale, a well-established instrument to assess work conditions during the past 12 months [43]. We formed the following three recognized subscales: Job Control, Cronbach’s alpha 0.82; Job Demand, alpha 0.70; and Job Support, alpha 0.75.

Cigarette smoking is associated with subsequent depression, and those suffering from depression are more likely to start smoking and become nicotine-dependent [36,44,45,46,47]. Smoking status was studied using the question “Which of the following alternatives describes your current smoking behavior the best?” with response choices of 1 ”I smoke daily 20 cigarettes or more”, 2 “I smoke daily 10 to 19 cigarettes”, 3 “I smoke daily 1 to 9 cigarettes”, 4 “I smoke once a week or more often, not daily however”, 5 “I smoke less frequently than once a week”, 6 “I have stopped or quit smoking”, 7 “I have tried smoking but I don’t smoke” and 8 “I have never even tried”. We collapsed the categories into current daily smokers, occasional smokers, former smokers and never smokers for the analyses

In youth, lower socio-economic status has been associated with higher rates of depressed mood and anxiety [48], and we used attained education as a measure of socio-economic status. During the data collection, in the Finnish educational system, compulsory education consisted of 9 years of basic school (ending at age 16). Sometimes an additional 10th year of the basic school was necessary. After that, upper secondary education was divided into vocational secondary education (non-academic) and academic secondary education (academic high school with matriculation examination), which normally took 2 and 3 years to complete, respectively. Tertiary-level education was provided by polytechnic schools (polytechnic degree) and universities (academic degree), lasting normally 3.5 and 5 years, respectively. Education was studied using the question “What schooling/degrees have you completed?” The response choices were 1 “Senior grades of the basic school”, 2 “I have a degree from the 10th class of the basic school“, 3 “I have passed my matriculation examination”, 4 “I have a vocational degree”, 5 “I have a polytechnic degree”, 6 “I have an academic degree”, 7 “I have passed an apprenticeship training”, and 8 “I have passed a vocational training or vocational course”. Based on these items, we classified the participants based on their secondary-level education as having no secondary education, having an academic high school diploma or having vocational training. Since many participants were still studying, differentiation into those with and those without a tertiary degree was not possible. In Finland, a tertiary degree can be obtained regardless of the kind of secondary degree.

Mental health assessments: Depressive symptoms were studied at age 17 (wave 3) and at age 22 (wave 4) using the General Behavior Inventory (GBI). The GBI is a self-reported inventory for mood-related behaviors that was first developed by Depue in 1981 and was designed to identify the presence and severity of depressive and manic or hypomanic symptoms and to assess for cyclothymia [49,50]. We used the short version of the GBI, which has been used in previous analyses of data from the FT12 sample [36,37,50,51]. This scale consists of 10 items inquiring about the occurrence of depressive symptoms, answered on a 4-point Likert scale from 0 = never to 3 = very often, and the total sum score ranges from 0–30. The coefficient alpha was 0.90 for GBI at wave 3 (age 17) and 0.91 at wave 4 (young adult), indicating excellent internal consistency for the scale.

Validation of the 10-item GBI at young adult age versus major depressive disorder DSM-IV diagnosis was performed. Of the respondents included in the analysis, 749 (17 (2.2%) missing) were also successfully interviewed, either in person or by telephone using the Semi-Structured Assessment for Genetics of Alcoholism psychiatric interview (Bucholz et al. 1994) to yield a diagnosis of lifetime Major Depressive Diagnosis based on DSM-IV [52]. A positive diagnosis was given by 86 participants (11.4% overall, among males 7.0% and among females 15.3%). In a logistic regression analysis, GBI at age 22 strongly predicted (*p* = 5.04 × 10^−23^, pseudo-R^2^ = 21.7%) a diagnosis of major depression, with an ROC (receiver operating characteristic) AUC (area under curve) of 0.831. This demonstrates that the short form of the GBI has excellent validity when measured against a psychiatric diagnosis of major depression, when obtained by trained and experienced interviewers using a validated and structured instrument.

Another assessment of depressive symptoms was obtained using the modified Multidimensional Peer Nomination Inventory (MPNI). Depressive symptoms were rated at ages 17 (wave 3) and 14 (wave 2). The adolescent rated him- or herself at ages 14 and 17. The MPNI includes 37 items covering major domains of personality and behavior. The core items of the MPNI were developed to represent a model of emotional and behavioral regulation. There are subscales of aggression, depression, hyperactivity/impulsivity, inattention, and social anxiety [53,54]. At age 14, there are five items of depression questions and, at age 17, two items of depression [55]. The ROC AUC for MPNI depressive symptoms at ages 14 against age 22 DSM-IV major depressive disorder was under 0.60, while the age 17 measures (both MPNI and GBI) had AUCs around 0.68, compared to 0.83 for age 22 GBI against structured interview-based major depressive disorder diagnosis. 

Appendix A consists of (1) the 17-year-old twin questionnaire including MPNI and GBI questions, and (2) the 21–24-year-old twin home questionnaire including work and noise questions.

Statistical analyses: Linear regression analyses were used to analyze the association of perceived noise at work with depressive symptoms. Regression coefficients (beta) and 95% confidence intervals (95% CIs) are reported. The main analysis was to examine the associations of perceived occupational noise exposure and depressive symptoms in young adulthood, both at age 22. We then examined whether depressive symptoms on the same scale (GBI) assessed at age 17 were associated with later perceived noise exposure at work at age 22. Noise exposure at work at age 17 would be minimal or absent, thus providing clues as to the direction of effects. Lastly, we used another measure of depressive symptoms (the MPNI assessed at ages 14 and 17) to see whether the GBI and MPNI work similarly at age 17 and whether early adolescence depressive symptoms show any association with perceived occupational noise exposure years later. Models with and without covariates were used to assess the statistical independence of the association. Further, we tested whether noise sensitivity and noise annoyance moderated the effect of perceived noise exposure through interaction analyses. All statistical analyses were conducted with Stata (version 17, StataCorp, College Station, TX, USA) [56].

## 3. Results

Table 1 shows the outcomes and covariates by perceived noise exposure categories (daily, weekly, occasionally, never). The participants were young adults and thus only a few had completed a university or polytechnic degree. A significant proportion of high school graduates continued to study. In contrast, only a small proportion of vocational school attendees were studying. We included those participants who had been working daily (*n* = 521) or weekly (*n* = 245) during the past 12 months in further analyses.

Males were exposed to noise more than females. Of those who were exposed to noise at work daily, 64.0% were male, of weekly exposed, 67.0% were male, and of occasionally exposed, 53.2% were male. Of those never exposed to noise at work, 68.6% were female (Table 1).

Of all participants, 32.1% were noise sensitive. Females were more noise sensitive than males (40.4% of females and 22.1% of males were noise sensitive). Overall, 51.9% of the participants reported noise annoyance at work, of whom 37.4% reported low noise annoyance, 8.8% moderate annoyance and 5.8% high annoyance at work. Males reported more annoyance than females, with 60.2% of males and 44.4% of females reporting noise annoyance at work. More males (7.5%) than females (4.2%) reported high noise annoyance at work.

Figure 1 presents the distribution of the GBI score at age 22 by sex and Figure 2 the distribution of the GBI score at age 22 by perceived noise exposure at work.

### 3.1. Association of Depressive Symptoms with Perceived Occupational Noise Exposure

Table 2 provides linear multivariable regression models that were used to investigate the association of depressive symptoms at age 22 with perceived occupational noise exposure as young adults (also at age 22). At age 22, in all participants in the age- and sex-adjusted model, daily noise exposure at work was associated with depressive symptoms (beta 1.56; 95% CI 0.54, 2.59), and also occasional noise exposure at work was associated with depressive symptoms (beta 1.17; 95% 0.43, 1.92), but the effect size of the association was smaller than in perceived daily noise exposure. In males, there was no association of depressive symptoms with noise exposure at work (beta 0.61; 95% CI −0.57, 1.78). In females, perceived daily noise exposure at work was associated with depressive symptoms (beta 2.75; 95% CI 0.98, 4.51) (Table 2).

In linear regression multivariable models adjusted for all non-noise covariates (age, sex, education, smoking, work conditions), in all participants, perceiving daily noise exposure at work was associated with depressive symptoms (beta 1.20; 95% CI 0.19, 2.20). In males and females separately, the point estimates were significant for females (beta 2.07; 95% 0.37, 3.76) but not for males (beta 0.42; 95% −0.82, 1.66). In the full model, noise sensitivity was independently associated with depressive symptoms in all participants (beta 1.35; 95% CI 0.54, 2.17) and in males (beta 1.96; 95% CI 0.68, 3.24), but not in females (beta 1.05; 95 % CI −0.04, 2.13) (Table 2). Noise sensitivity was not associated with perceived occupational noise exposure at work (results not included in tables).

In a full linear regression model adjusted for all variables, in all participants, daily noise exposure at work was associated with depressive symptoms (beta 1.19; 95% CI 0.09; 2.29). In males, there was no association of depressive symptoms with noise exposure at work (beta 0.22; 95% CI −1.08, 1.52), whereas in females, perceived daily noise exposure at work was associated with depressive symptoms (beta 2.22; 95% CI 0.34, 4.09). (Table 2).

As an additional analysis, we also ran the multivariable GBI model at age 22 by strata of education (none, vocational and academic high school) for females, and the association of GBI at age 22 and perceived occupational noise exposure, controlled for all covariates, was as strong in all subgroups.

In post hoc analyses, we looked at the overall interaction test between four categories of perceived noise exposure and two categories of noise sensitivity in females. It was not significant (*p* = 0.275). It is interesting that, among the seven combinations of exposure and sensitivity compared to non-exposed, non-sensitive females, females who were both exposed to daily noise and who were noise sensitive had more depressive symptoms than the non-sensitive ones (nominal *p* = 0.037; beta 3.47; 95% CI 0.21, 6.75).

Noise annoyance and perceived occupational noise exposure were highly correlated (rho = −0.71). In a linear regression multivariable model adjusted for perceived occupational noise exposure at work, education, smoking, and work conditions, annoyance was not associated with depressive symptoms in all participants or in males or females separately (tables not included). Noise annoyance was associated with perceived occupational noise exposure but not with depression, and statistical adjustment for annoyance did not remove the association of perceived occupational noise exposure with depression.

#### 3.1.1. Pre-Existing Depressive Symptoms (Using GBI) at Age 17 and Perceived Noise Exposure at Work as Young Adults

We also studied the association of pre-existing depression assessed using the GBI at age 17 with the perceived occupational noise exposure at age 22. Table 3 shows linear multivariable regression models that were used to investigate the association of pre-existing depressive symptoms with perceived occupational noise exposure. In all participants, in the age- and sex-adjusted model, perceived daily noise exposure at work (beta 1.69; 95% CI 0.45, 2.92) and occasional noise exposure at work (beta 1.10; 95% 0.19, 2.01) were associated with pre-existing depressive symptoms; the effect size of the association of perceived occasional noise exposure was smaller than in perceived daily noise exposure. In females, in the age- and sex-adjusted model, daily noise exposure at work (at age 22) was associated with pre-existing depressive symptoms (beta 3.02; 95% CI 0.70, 5.33), but in males there was no association (beta 0.53; 95% CI −0.72, 1.78) (Table 3).

In linear regression models adjusted for all non-noise covariates, in all participants, perceiving daily noise exposure at work was associated with depressive symptoms at age 17 (beta 1.36; 95% CI 0.12, 2.60). In males and females separately, the point estimates were higher for females (beta 2.18; 95% −0.15, 4.51) than for males (beta 0.61; 95% −0.65, 1.86), although neither was statistically significant as the sample sizes decreased (Table 3).

In a full model adjusted for all variables, in all participants, perceived daily noise exposure at work (at age 22) was associated with depressive symptoms at age 17 (beta 1.32; 95% CI 0.11; 2.53). In males, there was no association of depressive symptoms with perceived noise exposure at work (beta 0.43; 95% CI −0.80, 1.67), whereas in females perceived daily noise exposure at work was associated with depressive symptoms (beta 2.41; 95% CI 0.12, 4.70). Noise sensitivity was independently associated with depressive symptoms in all participants (beta 2.11; 95% CI 1.25, 2.97), and in males (beta 1.57; 95% CI 0.41, 2.74) and females (beta 2.43; 95 % CI 1.24, 3.62) (Table 3).

#### 3.1.2. Pre-Existing Depressive Symptoms (Using MPNI) at Age 14 and 17 and Perceived Noise Exposure at Work as Young Adults

We also studied pre-existing depressive symptoms using the MPNI. Depressive symptoms were rated at ages 14 and 17. Table A1 and Table A2 present the age 14 and age 17 self-rated MPNI depressive symptoms regression analyses against perceived noise exposure at work using the same models as for GBI at age 17 in Table 3. While the regression coefficient for perceived daily noise exposure at work (at age 22) was the highest in almost all models, none were statistically significant among all participants (Table A1 and Table A2). Only among males at the age of 14 in Model b, self-rated MPNI depressive symptoms were associated with perceived daily noise exposure at work at age 22 (beta 0.11; 95% CI 0.01, 0.22) (Table A1) and among males at the age of 17 in Model c (full model), self-rated MPNI depressive symptoms were associated with perceived occasional noise exposure at work at age 22 (beta 0.17; 95% CI 0.00, 0.34) (Table A2). While the regression coefficient for perceived daily noise exposure (at age 22) was in almost all models the highest, none were statistically significant (in contrast to GBI at age 17).

## 4. Discussion

In the current study, we investigated the association of perceived occupational noise exposure with depressive symptoms in young adults. In all participants and in females, perceived daily noise exposure at work was associated with depressive symptoms. In contrast, among males there was no association of depressive symptoms with perceived noise exposure at work. In both males and females, depressive symptoms were associated with noise sensitivity, but not with noise annoyance in young adulthood.

The analysis in young adulthood was cross-sectional and hence we cannot provide evidence for the direction of association, i.e., whether depression is leading to greater perceived noise exposure or whether perceived noise exposure increases symptoms of depression. To this end, we also studied the pre-existing depressive symptoms at ages 17 using GBI (and MPNI at ages 14 and 17).

GBI at age 17 was consistently associated with noise exposure at age 22 overall and among females, even after adjustment for all covariates. Thus, the direction of the effect from age 17 would be from depressive symptoms to perceived noise exposure. It is possible that the association of prior depression with working in noisier occupations could be explained by a healthy worker selection effect. Those with depressive symptoms may also report the same noise levels as louder than those without depressive symptoms. The association of depression symptoms assessed using the MPNI at ages 14 and 17 showed weaker and less consistent effects than for GBI. As the perceived occupational noise exposure was reported only at age 22, we had no data on noise exposure at work for ages 14 or 17, but it would be very unlikely that the participants would have occupational noise exposure at those ages. Depressive symptoms at age 14 showed no association, perhaps due to changes in symptoms over time as the young adolescent matures. We compared self-ratings of GBI at age 17 to MPNI at age 17 to see whether the scales work consistently, and self-ratings of MPNI age 17 to age 14 to see age effects. The measures were reasonably correlated, but the MPNI depression scales are based on fewer items, so, psychometrically, MPNI is not as powerful as GBI. Thus, the lack of significance in age 17 MPNI depression is not surprising, while the predictive power (and correlations from age 14) are quite weak.

Sex differences in the effects of occupational noise have also been found in previous studies [57,58]. The study of Abbasi et al. (2022) investigating sex differences in cognitive performance and psychophysiological responses during noise exposure on tasks with different workloads found that females and males indicate significant and different responses in exposure to different noise levels and workloads. Female participants reported significantly higher annoyance levels and fatigue than males. Females had better cognitive performance at sound levels of < 65 dBA with a low and medium mental workload, while males had better cognitive performance at sound levels > 65 dB and a high workload. Noise-induced stress effects in females were more pronounced than in males [57]. In a Brazilian study by Oenning et al. (2018), exposure to noise and chemicals, workplace violence, and intense physical activity were risk factors for major depressive disorder in females, and prolonged exposure to sun in males. Risk factors for both males and females were stress at work and working part-time [58]. In our study, perceived daily noise exposure at work was associated with depressive symptoms in females only. Females may be more vulnerable to the health effects of occupational noise than males or they may be more aware of noise exposure.

In the present study, perceived noise exposure and noise sensitivity were independently associated with depressive symptoms in all participants and in females. Sandrock et al. (2008) have previously studied mental strain in noise sensitive persons working under moderate levels of noise. Significant effects for noise sensitivity and the kind of task being performed were found. In that study, noise sensitive persons evaluated noisy situations as more annoying and experienced a higher level of strain than noise insensitive persons. Noise sensitivity had a relationship with emotional stress. [59]. In a population-based study conducted in two large cities in South Korea by Lim et al. (2018), it was found that noise sensitivity is significantly associated with internalizing, externalizing, and total behavioral problems. Noise exposure was associated with total behavioral problems [60]. 

In this study, noise sensitivity was independent of perceived occupational noise exposure. We did not perform any moderator/mediator analyses. In previous studies, noise sensitivity has also been independent of noise exposure levels indicated in transportation noise maps [39,40,61].

In the present study, noise sensitivity was associated with depressive symptoms in both males and females, which is consistent with its more biological basis as a psychological trait. In an EEG/MEG study by Kliuchko et al. (2016), noise sensitivity was specifically related to neural mechanisms linked to the processing of noise [62]. Noise sensitivity has its origins in primary auditory functions of the central nervous system [62]. An MRI study by Kliuchko et al. (2018) found that individual differences in noise sensitivity are associated with the structural organization of brain areas playing a role in auditory perception, interoception, and the processing of emotions and salience [63]. In this context, our intriguing post hoc analysis result was the observation that, among females reporting daily occupational noise, the noise sensitive individuals have more depressive symptoms than the non-sensitive ones. The association in this subgroup may be a chance finding, due to multiple testing within the overall interaction analysis, which was not statistically significant. This would need to be independently replicated in another study of young adults.

We had only self-reported data on noise exposure at work, which is a major limitation of this study. We asked the participants if they had been exposed to such a loud noise at work that they could not hear normal speech within a meter’s distance, which means that the sound level was exceeding 60 dB, as this is the sound level of normal conversation [64]. The hearing of the participants was not assessed. We need further longitudinal studies on the association of measured or modeled occupational noise with depression. The strengths of this study are that it is population-based and that participants were at the beginning of their careers and had not changed their workplace many times.

## 5. Conclusions

In young adults, females, but not males, perceived occupational noise exposure was associated with depressive symptoms. Noise sensitivity was independently associated with depressive symptoms among all, and separately for males but not females. Noise sensitivity was independent of perceived occupational noise exposure. Measures of depressive symptoms in late adolescence (age 17) were predictive of noise exposure, suggesting complex interactions of noise and depression in adolescents and young adults.

## Figures and Tables

**Figure 1 ijerph-20-04850-f001:**
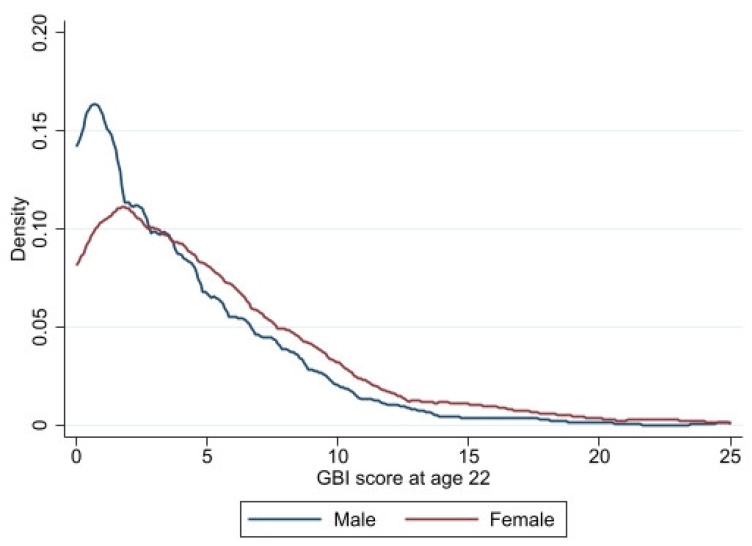
Distribution of the GBI score at age 22 by sex.

**Figure 2 ijerph-20-04850-f002:**
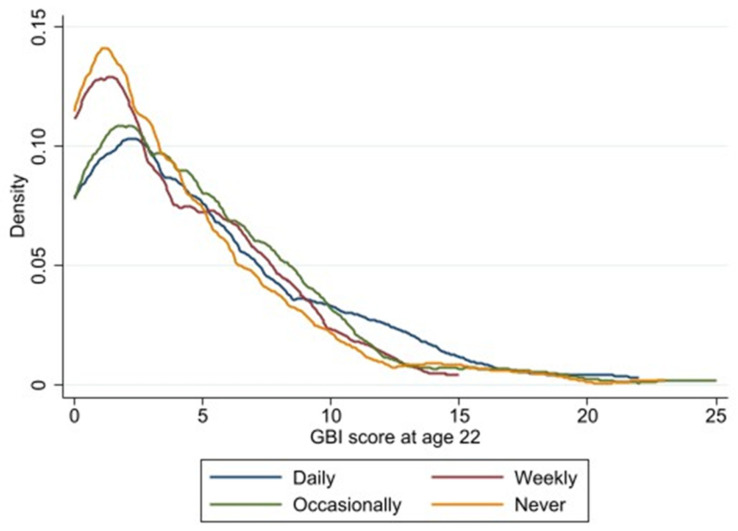
Distribution of the GBI score at age 22 by perceived noise exposure at work.

**Table 1 ijerph-20-04850-t001:** Distributions of variables (outcomes and covariates) by noise exposure categories in young adulthood among participants who had worked at least weekly in the prior 12 months.

Perceived Noise Exposure at Work					
	Daily	Weekly	Occasionally	Never	Total
	*N* = 114	*N* = 97	*N* = 233	*N* = 322	*N* = 766
Sex (N, %)					
Male	73 (64.0%)	65 (67.0%)	124 (53.2%)	101 (31.4%)	363 (47.4%)
Female	41 (36.0%)	32 (33.0%)	109 (46.8%)	221 (68.6%)	403 (52.6%)
Age at wave 4 (years) (Mean, SD)	22.4 (0.7)	22.3 (0.7)	22.5 (0.7)	22.5 (0.8)	22.4 (0.7)
Secondary education (N, %)					
None	7 (6.1%)	9 (9.3%)	21 (9.0%)	11 (3.4%)	48 (6.3%)
Vocational	69 (60.5%)	44 (45.4%)	94 (40.3%)	93 (28.9%)	300 (39.2%)
Academic	38 (33.3%)	44 (45.4%)	118 (50.6%)	218 (67.7%)	418 (54.6%)
Smoking status (N, %)					
Never	43 (37.7%)	37 (38.1%)	85 (36.6%)	153 (47.5%)	318 (41.6%)
Former	15 (13.2%)	13 (13.4%)	22 (9.5%)	30 (9.3%)	80 (10.5%)
Occasional	6 (5.3%)	7 (7.2%)	28 (12.1%)	40 (12.4%)	81 (10.6%)
Current	50 (43.9%)	40 (41.2%)	97 (41.8%)	99 (30.7%)	286 (37.4%)
Work conditions (Mean, SD)					
Job support	2.2 (0.9)	2.0 (0.8)	2.0 (0.7)	2.0 (0.8)	2.0 (0.8)
Job demand	0.3 (0.7)	0.3 (0.7)	0.4 (0.7)	0.6 (0.7)	0.4 (0.7)
Job control	1.5 (0.9)	1.4 (0.7)	1.5 (0.8)	1.6 (0.8)	1.5 (0.8)
Current study (N, %)					
No	83 (73.5%)	70 (72.9%)	131 (57.7%)	147 (45.7%)	431 (56.9%)
Only study	14 (12.4%)	8 (8.3%)	24 (10.6%)	38 (11.8%)	84 (11.1%)
Study&work	16 (14.2%)	18 (18.8%)	72 (31.7%)	137 (42.5%)	243 (32.1%)
GBI for age 22 (Mean, SD)	5.0 (4.8)	3.6 (3.5)	4.8 (4.5)	3.9 (4.2)	4.3 (4.3)
GBI for age 17 (Mean, SD)	5.5 (5.4)	4.1 (4.1)	5.3 (5.2)	4.7 (4.8)	4.9 (5.0)

**Table 2 ijerph-20-04850-t002:** Linear multivariable regression models of the association of depressive symptoms at age 22 using the GBI scale with perceived occupational noise exposure and selected covariates in young adulthood.

GBI Score at Age 22	Regression Coefficient Beta (95% CI)								
	All			Male			Female		
	Model a (*n* = 757)	Model b (*n* = 756)	Model c (*n* = 658)	Model a (*n* = 360)	Model b (*n* = 360)	Model c (*n* = 302)	Model a (*n* = 397)	Model b (*n* = 396)	Model c (*n* = 356)
R^2^ for the model	0.0468	0.1134	0.1362	0.0253	0.0826	0.1315	0.0360	0.131	0.1463
Perceived noise at work									
Never	Ref.	Ref.	Ref.	Ref.	Ref.	Ref.	Ref.	Ref.	Ref.
Daily	1.56 (0.54, 2.59) *	1.20 (0.19, 2.20) *	1.19 (0.09, 2.29) *	0.61 (−0.57, 1.78)	0.42 (−0.82, 1.66)	0.22 (−1.08, 1.52)	2.75 (0.98, 4.51) *	2.07 (0.37, 3.76) *	2.22 (0.34, 4.09) *
Weekly	0.30 (−0.55, 1.16)	0.06 (−0.78, 0.89)	−0.12 (−1.05, 0.82)	−0.25 (−1.28, 0.77)	−0.39 (−1.39, 0.61)	−0.44 (−1.55, 0.66)	0.88 (−0.63, 2.39)	0.42 (−0.99, 1.82)	−0.02 (−1.58, 1.54)
Occasionally	1.17 (0.43, 1.92) *	0.94 (0.22, 1.66) *	1.01 (0.24, 1.77)	0.84 (−0.20, 1.88)	0.67 (−0.40, 1.75)	1.03 (−0.13, 2.18)	1.25 (0.19, 2.32)	1.01 (0.01, 2.00) *	0.82 (−0.25, 1.89)
Age at 22 years survey	0.28 (−0.16, 0.71)	0.26 (−0.16, 0.68)	0.14 (−0.33, 0.60)	0.54 (−0.08, 1.16)	0.43 (−0.18, 1.05)	0.29 (−0.41, 0.99)	0.03 (−0.56, 0.62)	0.12 (−0.46, 0.70)	0.03 (−0.61, 0.67)
Secondary education									
Academic		Ref.	Ref.		Ref.	Ref.		Ref.	Ref.
None		1.67 (0.13, 3.21) *	1.69 (−0.01, 3.39)		1.28 (−0.59, 3.15)	1.38 (−0.58, 3.34)		1.92 (−0.84, 4.68)	2.17 (−0.97, 5.31)
Vocational		−0.16 (−0.89, 0.58)	−0.22 (−1.03, 0.59)		−0.28 (−1.22, 0.67)	−0.21 (−1.25, 0.84)		−0.12 (−1.20, 0.96)	−0.17 (−1.32, 0.98)
Smoking									
Never		Ref.	Ref.		Ref.	Ref.		Ref.	Ref.
Former		1.62 (0.42, 2.82) *	1.46 (0.11, 2.81) *		1.27 (−0.30, 2.85)	1.04 (−0.65, 2.74)		1.81 (−0.00, 3.63)	1.90 (−0.26, 4.05)
Occasional		0.03 (−0.82, 0.89)	0.06 (−0.84, 0.97)		−1.40 (−2.43, −0.37) *	−1.59 (−2.69, −0.49) *		1.04 (−0.20, 2.29)	1.05 (−0.23, 2.33)
Current		1.50 (0.75, 2.26) *	1.44 (0.58, 2.30) *		0.72 (−0.30, 1.75)	0.53 (−0.64, 1.71)		2.13 (1.02, 3.23) *	2.11 (0.89, 3.32) *
Work conditions									
Job support		0.56 (0.14, 0.98) *	0.75 (0.28, 1.23) *		0.06 (−0.50, 0.62)	0.32 (−0.37, 1.01)		0.86 (0.29, 1.43) *	0.96 (0.33, 1.58) *
Job demand		−0.15 (−0.63, 0.32)	−0.15 (−0.67, 0.37)		−0.05 (−0.90, 0.80)	−0.05 (−1.00, 0.90)		−0.21 (−0.78, 0.35)	−0.18 (−0.80, 0.44)
Job control		0.48 (0.01, 0.95) *	0.36 (−0.15, 0.86)		0.54 (−0.02, 1.10)	0.34 (−0.28, 0.97)		0.43 (−0.27, 1.13)	0.36 (−0.37, 1.09)
Noise sensitivity									
No			Ref.			Ref.			Ref.
Yes			1.35 (0.54, 2.17) *			1.96 (0.68, 3.24) *			1.05 (−0.04, 2.13)

a: Independent variable: noise exposure at work, age, and sex; b: Independent variable: based on a, additional secondary level school, smoking status, job support, demand, and control; c: Independent variable: based on b, additional noise sensitivity; * *p* < 0.05.

**Table 3 ijerph-20-04850-t003:** Linear multivariable regression models of the association of depressive symptoms at age 17 using the GBI scale with perceived occupational noise exposure and selected covariates in young adulthood.

GBI Score at Age 17	Regression Coefficient Beta (95% CI)								
	All			Male			Female		
	Model a (*n* = 664)	Model b (*n* = 663)	Model c (*n* = 662)	Model a (*n* = 302)	Model b (*n* = 302)	Model c (*n* = 301)	Model a (*n* = 362)	Model b (*n* = 361)	Model c (*n* = 361)
R^2^ for the model	0.0780	0.1221	0.1584	0.0111	0.1115	0.1401	0.0295	0.1157	0.1597
Perceived noise at work									
Never	Ref.	Ref.	Ref.	Ref.	Ref.	Ref.	Ref.	Ref.	Ref.
Daily	1.69 (0.45, 2.92) *	1.36 (0.12, 2.60) *	1.32 (0.11, 2.53) *	0.53 (−0.72, 1.78)	0.61 (−0.65, 1.86)	0.43 (−0.80, 1.67)	3.02 (0.70, 5.33) *	2.18 (−0.15, 4.51)	2.41 (0.12, 4.70) *
Weekly	0.31 (−0.78, 1.40)	−0.13 (−1.30, 1.04)	0.04 (−1.08, 1.16)	−0.37 (−1.67, 0.94)	−0.31 (−1.69, 1.08)	−0.20 (−1.55, 1.15)	0.85 (−1.07, 2.77)	0.06 (−1.98, 2.09)	0.29 (−1.68, 2.27)
Occasionally	1.10 (0.19, 2.01) *	0.78 (−0.12, 1.68)	0.73 (−0.15, 1.62)	0.68 (−0.42, 1.79)	0.98 (−0.19, 2.15)	0.95 (−0.19, 2.10)	1.28 (−0.09, 2.66)	0.57 (−0.69, 1.84)	0.48 (−0.78, 1.73)
Age at 17 years survey	0.09 (−0.47, 0.66)	0.04 (−0.51, 0.58)	0.07 (−0.48, 0.61)	−0.07 (−0.69, 0.56)	−0.18 (−0.82, 0.46)	−0.18 (−0.80, 0.44)	0.21 (−0.61, 1.03)	0.08 (−0.71, 0.88)	0.12 (−0.69. 0.93)
Secondary education									
Academic		Ref.	Ref.		Ref.	Ref.		Ref.	Ref.
None		1.87 (−0.42, 4.17)	1.95 (−0.32, 4.22)		−0.83 (−2.54, 0.87)	−0.66 (−2.37, 1.04)		4.83 (0.05, 9.61) *	4.68 (−0.07, 9.42)
Vocational		−0.55 (−1.40, 0.31)	−0.50 (−1.34, 0.33)		−1.38 (−2.31, −0.44) *	−1.25 (−2.17, −0.32) *		0.23 (−1.07, 1.54)	0.17 (−1.10, 1.44)
Smoking									
Never		Ref.	Ref.		Ref.	Ref.		Ref.	Ref.
Former		1.85 (0.45, 3.26) *	1.33 (−0.06, 2.73)		1.44 (−0.19, 3.06)	1.16 (−0.42, 2.74)		1.66 (−0.73, 4.05)	0.91 (−1.49, 3.31)
Occasional		1.20 (−0.04, 2.44)	1.22 (−0.02, 2.47)		−1.42 (−2.76, −0.09) *	−1.51 (−2.90, −0.11) *		2.60 (0.84, 4.36) *	2.73 (1.01, 4.45) *
Current		1.07 (0.17, 1.97) *	0.96 (0.07, 1.86) *		0.29 (−0.72, 1.29)	0.28 (−0.72, 1.28)		1.51 (0.11, 2.91) *	1.28 (−0.12, 2.68)
Work conditions									
Job support		0.61 (0.09, 1.14) *	0.62 (0.11, 1.12) *		1.01 (0.29, 1.74) *	0.98 (0.26, 1.71) *		0.38 (−0.28, 1.05)	0.41 (−0.24, 1.05)
Job demand		−0.45 (−1.05, 0.14)	−0.44 (−1.03, 0.15)		0.19 (−0.61, 0.98)	0.18 (−0.62, 0.97)		−0.84 (−1.66, −0.03)	−0.81 (−1.62, −0.01) *
Job control		−0.24 (−0.77, 0.29)	−0.26 (−0.78, 0.26)		−0.06 (−0.67, 0.55)	−0.05 (−0.67, 0.56)		−0.39 (−1.18, 0.40)	−0.45 (−1.22, 0.31)
Noise sensitivity									
No			Ref.			Ref.			Ref.
Yes			2.11 (1.25, 2.97) *			1.57 (0.41, 2.74) *			2.43 (1.24, 3.62)

a: Independent variable: noise exposure at work, age, and sex; b: Independent variable: based on a, additional secondary level school, smoking status, job support, demand, and control; c: Independent variable: based on b, additional noise sensitivity; * *p* < 0.05.

## Data Availability

The FT12 data are not publicly available due to the restrictions of informed consent. However, the FT12 data are available through the Institute for Molecular Medicine Finland (FIMM) Data Access Committee (DAC) (fimm-dac@helsinki.fi) for authorized researchers who have IRB/ethics approval and an institutionally approved study plan. To ensure the protection of privacy and compliance with national data protection legislation, a data use/transfer agreement is needed, the content and specific clauses of which will depend on the nature of the requested data.

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
