# Peer review of "Perceived Occupational Noise Exposure and Depression in Young Finnish Adults"

_ijerph, 2023, doi:10.3390/ijerph20064850_

Round 1
Reviewer 1 Report
I congratulate the authors for the excellent work related to this article. In the article they evaluate the association between perceived occupational noise exposure and depressive symptoms in young Finnish adults and whether noise sensitivity moderates this association. The study included those who worked daily or weekly. The research described has the potential to become a good paper, but it needs to address a minor review, as noted below.
Lines 164-167: The explanation of the methods is mixed with the results. I would suggest splitting clearly the methodology from the results, in order to make the reading easier.
Lines 197-198. I would also suggest explaining more clearly how the coefficient alpha was obtained. If you do not consider it so important to be in the main text, you can add the results as supplementary materials or in one/several appendixes as needed.
Line 233. You say that you conduct Factor regression analysis. Where are the results of this analysis?
Line 234. You say that you conduct linear regression analysis. Do you mean linear regression or multivariate regression? If you mean linear regression analysis, where are the results of this analysis? Please, if necessary, change the abstract, in which you use also the expression linear regression.
Line 277: What does beta mean in your analysis? Please, explain in the text
In the author contribution section, I would suggest to use abbreviation of the names and surnames of the authors, following the criteria of the authors's guide of the IJERPH: https://www.mdpi.com/1660-4601/20/1/ 709
General comment: The paper contains no graphics at all. Graphics make reading easier, and sometimes can contain more information than plane text. They also allow the readers to take their own conclusions. If you have conducted factor analysis, for example, you could add a scree plot to evaluate the factors that explain most of the variability in the data, a score plot…
Author Response
Response to Reviewer 1 Comments
We are very grateful for the reviewer’s valuable comments and we have made the required changes in the manuscript.
Point 1: Lines 164-167: The explanation of the methods is mixed with the results. I would suggest splitting clearly the methodology from the results, in order to make the reading easier.
Response 1: As a methodological assessment, we carried out a factor analysis of the items in the Karasek scale. This was to ensure that our data set conforms to earlier studies in recognizing the three component scales. As our data was consistent with earlier work, we formed the scales by summing the items contributing to each subscale. The Cronbach alpha is a commonly used reliability coefficient of the internal consistency of a scale. As we are describing the measures we use, the alpha values are not a result but a descriptive of the measure and belongs to the methods section.
In the text, we have now deleted the mention of the factor analysis as it is not essential (lines 167 and 233).
Point 2: Lines 197-198. I would also suggest explaining more clearly how the coefficient alpha was obtained. If you do not consider it so important to be in the main text, you can add the results as supplementary materials or in one/several appendixes as needed.
Response 2: Please see our response above.
Point 3: Line 233. You say that you conduct Factor regression analysis. Where are the results of this analysis?
Response 3: As described above, the factor analysis was a secondary analysis we conducted to ensure that the Karasek scales behaved as expected. Because the factor analysis did not provide any new information, we scored the Karasek subscales in accordance with prior practice. Accordingly, we delete the first sentence of the Statistical analyses (line 233).
Point 4: Line 234. You say that you conduct linear regression analysis. Do you mean linear regression or multivariate regression? If you mean linear regression analysis, where are the results of this analysis? Please, if necessary, change the abstract, in which you use also the expression linear regression.
Response 4: We thank the reviewer for pointing this out. We did use linear regression models, but did not indicate that clearly in the Table headings and text. We have now clarified the terminology to indicate that Linear multivariable regression models were used, where the dependent variable (such as GBI) is a continuous outcome and there are one or more independent variables (hence multivariable). Table 2., 3., A1 and A2 headings have been changed (Linear multivariable regression models ...).
Point 5: Line 277: What does beta mean in your analysis? Please, explain in the text
Response 5: Beta is the regression coefficient for each independent variable in the model, which provide an estimate of the change in the dependent variable per unit change in the independent variable. We have clarified this in the Table 2.,3.,A1 and A2 headers (Instead of Beta now Regression Coefficient Beta) and added a sentence in Statistical analyses on lines 234-235: "Regression coefficients (beta) and 95% confidence intervals (95% CIs) are reported."
Point 5: In the author contribution section, I would suggest to use abbreviation of the names and surnames of the authors, following the criteria of the authors's guide of the IJERPH: https://www.mdpi.com/1660-4601/20/1/ 709
Response 5: Required changes have been made. We’ve used abbreviation of the names and surnames of the authors.
Point 6: General comment: The paper contains no graphics at all. Graphics make reading easier, and sometimes can contain more information than plane text. They also allow the readers to take their own conclusions. If you have conducted factor analysis, for example, you could add a scree plot to evaluate the factors that explain most of the variability in the data, a score plot…
Response 5: We have added two figures: Figure 1 presents the distribution of the GBI score at age 22 by sex and Figure 2 the distribution of the GBI score at age 22 by perceived noise exposure at work.
Reviewer 2 Report
Did the authors previously establish that the subjects had a correct hearing? When and how, if they did? If not, please state it in the limiting factors.
Deaf and hard of hearing people will not hear the interlocutor's voice at a meter distance even when there is no noise.
Interesting article. Kudos to the authors.
Author Response
Response to Reviewer 1 Comments
We are very grateful for the reviewer’s valuable comments and we have made the required changes in the manuscript.
Point 1. Did the authors previously establish that the subjects had a correct hearing? When and how, if they did? If not, please state it in the limiting factors.
Deaf and hard of hearing people will not hear the interlocutor's voice at a meter distance even when there is no noise.
Response 1. We have not assessed the hearing of the participants. As young adults representing the general population, the prevalence of impaired hearing is likely to be low. We have added a sentence in the limitations that the hearing of the participants was not assessed (line 461).